Learning to perform role-filler binding with schematic knowledge

Chen Catherine 1 cc27@alumni.princeton.edu
http://orcid.org/0000-0002-0730-5240 Lu Qihong 2
Beukers Andre 2
http://orcid.org/0000-0003-3540-5019 Baldassano Christopher 3
http://orcid.org/0000-0002-5887-9682 Norman Kenneth A. 2
1 Department of Computer Science, Princeton University , Princeton, NJ , USA
2 Department of Psychology, Princeton University , Princeton, NJ , USA
3 Department of Psychology, Columbia University , New York, NY , USA
Gollo Leonardo
Electronic publication date: 2021 Mar 31
Publication date: 2021
Volume: 9
Electronic Location ID: e11046
Received 2019 Nov 5; Accepted 2021 Feb 10
Copyright: © 2021 Chen et al.
Copyright year: 2021
Copyright holder: Chen et al.
License: This is an open access article distributed under the terms of the Creative Commons Attribution License, which permits unrestricted use, distribution, reproduction and adaptation in any medium and for any purpose provided that it is properly attributed. For attribution, the original author(s), title, publication source (PeerJ) and either DOI or URL of the article must be cited.
License URL: https://creativecommons.org/licenses/by/4.0/

Keywords: Schema, Schemata, Neural networks, Frames, Roles, Fillers, Role-filler binding

Funding: Intel Labs and Multi-University Research Initiative ONR/DoD N00014-17-1-2961 Funding was provided by a grant from Intel Labs and by a Multi-University Research Initiative grant to Kenneth A. Norman (ONR/DoD N00014-17-1-2961). The funders had no role in study design, data collection and analysis, decision to publish, or preparation of the manuscript.

==============================
Through specific experiences, humans learn the relationships that underlie the structure of events in the world. Schema theory suggests that we organize this information in mental frameworks called “schemata,” which represent our knowledge of the structure of the world. Generalizing knowledge of structural relationships to new situations requires role-filler binding, the ability to associate specific “fillers” with abstract “roles.” For instance, when we hear the sentence Alice ordered a tea from Bob, the role-filler bindings customer:Alice, drink:tea and barista:Bob allow us to understand and make inferences about the sentence. We can perform these bindings for arbitrary fillers—we understand this sentence even if we have never heard the names Alice, tea, or Bob before. In this work, we define a model as capable of performing role-filler binding if it can recall arbitrary fillers corresponding to a specified role, even when these pairings violate correlations seen during training. Previous work found that models can learn this ability when explicitly told what the roles and fillers are, or when given fillers seen during training. We show that networks with external memory learn to bind roles to arbitrary fillers, without explicitly labeled role-filler pairs. We further show that they can perform these bindings on role-filler pairs that violate correlations seen during training, while retaining knowledge of training correlations. We apply analyses inspired by neural decoding to interpret what the networks have learned.

Introduction

As humans, we have a powerful ability to learn the relationships underlying the structure of events, and to use them to organize and guide cognition. Knowing how events are structured in the world allows humans to understand and interact with novel situations. Schema theory suggests that mental frameworks called “schemata” organize our knowledge of the world. Humans learn schemata through experiences and use them as building blocks for understanding the world. For example, we learn the schema for “visiting coffee shops” based on individual experiences at specific coffee shops. Although each coffee shop visit differs slightly from the others, the experiences share some underlying structure, which we are able to learn without being explicitly instructed about the underlying structural relationships. Previous work studying these phenomena has referred to these structures as “scripts” (e.g., Schank & Abelson, 1977; Bower, Black & Turner, 1979; Miikkulainen & Dyer, 1991) and “frames” (e.g., Minsky, 1974; Brachman & Schmolze, 1985), and has shown that they affect people’s memory of events (Frederic Charles Bartlett, 1932; Bower, Black & Turner, 1979).

These structures can be viewed as frames consisting of abstract “roles” which are occupied by specific “fillers” (Minsky, 1974; Brachman & Schmolze, 1985). For instance, a schema for coffee shops might include the roles barista, drink, and customer. Knowing the coffee shop schema allows us to understand and make inferences based on the sentence Alice ordered a tea from Bob, by inferring the role-filler relations barista:Bob, drink:tea, customer:Alice. We can do this even if we have no idea what the words Alice, tea, and Bob mean. This kind of inferential process critically relies on an operation that binds a specific filler (e.g., Alice) to a known structural “role” (e.g., customer). This process is commonly referred to as “role-filler binding”. Role-filler binding is essential for understanding and organizing structural relationships within the world, allowing us to learn flexible, composable building blocks with which we can understand new situations.

Role-filler binding involves the ability to systematically apply propositions to new fillers. It allows us to generalize schematic knowledge to novel instances, by applying the relationships in a schema to arbitrary fillers. For instance, if we understand the phrase Alice ordered a tea from Bob, we should be able to understand the phrase A ordered a C from B for any fillers A, B, and C. Previous work suggests two important features that can allow systems to perform complex compositional reasoning: dynamic binding (the ability to bind the same filler to different roles), and independent binding (using representations in which bindings are independent of bound arguments) (Holyoak & Hummel, 2000).

In this work we focus on learning to extract role-filler bindings from sentence structure. We create a task in which role-filler bindings are identified based on the structure of input sentences, and test whether networks learn to recall role-filler pairs based on an input sentence. We test whether networks can perform this task for arbitrary role-filler pairs, including those that violate role-filler correlations seen during training, while still remembering those correlations. We identify a training regime and model that satisfy these criteria.

We characterize architecture- and task- boundaries of this ability by identifying models that are not sufficient for role-filler binding. We also identify test inputs with structures that heavily diverge from those seen during training, that our model and training combinations do not solve.

Specifically, we show that connectionist architectures with external memory (the Fast Weights and Differentiable Neural Computer architectures) are able to perform role-filler binding without explicitly labeled roles and fillers, and that they can generalize to arbitrary fillers if they see a sufficiently diverse set of training examples. We find that if we impose statistical regularities between roles and their corresponding filler distributions, the Fast Weights network displays knowledge of these correlations, while still generalizing to role-filler pairs that violate these regularities. Lastly, we provide additional analyses inspired by neural decoding that give insights into how the models solve this task.

Related work

Previous work pointed to the importance of being able to represent role-filler bindings independently and dynamically, such that roles can be bound to different fillers in different scenarios (e.g., Hummel & Biederman, 1992; Doumas & Hummel, 2005). Traditional connectionist models that achieve role-filler binding through conjunctive coding are unable to dynamically represent these symbolic relations, because of an inability to separate the representations of role-filler bindings from the representations of the fillers themselves (e.g., Fodor & Pylyshyn, 1988; Doumas & Hummel, 2005).

Prior work has shown that neural networks can learn role-filler binding if they are given inputs that explicitly provide the role corresponding to each filler, or if they are given train and test inputs that share the same pool of fillers. For instance, some models use specific input units to encode roles such as the action and agent of the example (Kriete et al., 2013; Elman & McRae, 2019), and others use holographic reduced representations (Plate, 1995) to explicitly encode the action, agent, and patient of the input examples (Franklin et al., 2019). Another line of work identified models that can perform role-filler binding when tested on examples containing the same pool of fillers during train and test (St. John & McClelland, 1990; Miikkulainen & Dyer, 1991; Hinaut & Dominey, 2013). In some cases, these models can generalize to unseen grammatical constructions that are compositions of trained constructions, if the examples contain the same set of fillers during train and test (Hinaut & Dominey, 2013). Others have tested relational role-filler binding in neural networks, testing models’ ability to perform role-filler binding on examples containing previously unseen fillers, role-filler bindings that violate statistical correlations seen during training, and input segments that are presented in a shuffled order (Puebla, Martin & Doumas, 2019). This work showed that a Story Gestalt (St. John & McClelland, 1990) and a Seq2Seq with Attention (Bahdanau, Cho & Bengio, 2015) model fail to perform role-filler binding when given novel fillers or bindings that violate correlations seen during training.

Other related work developed a model that learns role-filler bindings based on the features of objects, in a manner related to behavioral observations of how children learn relations (Doumas, Hummel & Sandhofer, 2008). The model begins with representations of objects as collections of features (for instance, apple could be represented by the collection of features (red, size-2, fruit)). It uses asynchronous firing sequences and comparator functions to learn representations of predicates (for instance, forming an explicit representation of the idea red based on seeing apple and firetruck), as well as relations between objects (for instance, learning that the respective features size-2 and size-5 for apple and firetruck imply the relation bigger(firetruck, apple)).

Our work builds upon these findings by identifying models that learn role-filler bindings, in a way that generalizes to novel fillers, and that also generalizes to fillers that violate correlations seen during training. In our setting, the networks must learn to perform role-filler binding based on the structure of the sentence. Rather than representing inputs as the conjunction of a filler’s features or using input representations that explicitly provide role-filler conjunctions, we focus on a setting in which input words are represented by randomly generated vectors.

Our decoding analyses are inspired by multivariate pattern analyses used to decode neural data (Norman et al., 2006). Rather than neuroimaging data, we apply these methods to neural network activity. Previous researchers have used various mapping methods to gain insight into neural network activity based on the activation of network layers. For instance, previous work has used stimulus-decoding analyses and activation similarity to probe for features represented by networks and to gain insight into the processing stages corresponding to layers of the networks (e.g., Ettinger, Elgohary & Resnik, 2016; Qian, Qiu & Huang, 2016; Hupkes, Veldhoen & Zuidema, 2018; Guest & Love, 2019; Lakretz et al., 2019; Tenney, Das & Pavlick, 2019).

Methods

Given a sentence such as Alice ordered a tea from Bob, humans know to extract the associations barista:Bob, drink:tea, customer:Alice. From this we can infer the relation served(Bob, Alice). This operation is dependent on the structure of the sentence—given the sentence Bob ordered a tea from Alice, we would instead extract the associations barista:Alice, drink:tea, customer:Bob and therefore infer the relation served(Alice, Bob). Furthermore, we are able to generalize this to arbitrary fillers—given the sentence Alice ordered a tea from Sam, we can extract the associations barista:Sam, drink:tea, customer:Alice even if we have never encountered Sam, or if Sam is usually the customer rather than the barista. Though we can extract arbitrary role-filler bindings, we also retain knowledge of the likely fillers for each role. If Alice is usually the customer, but an input sentence explicitly states that Sam is the customer, we know to extract customer:Sam in this situation. However, given an ambiguous input sentence such as BLANK ordered a tea from Bob, we would tend to guess customer:Alice. In the remainder of this section, we describe experiments designed to test these behaviors in neural network models.

We represented a schema as an underlying graph that defines story states and transition probabilities between those states, and we produced stochastically generated stories based on the schema. Each state includes fixed frame-text and variable roles, and the roles are substituted with fillers drawn from a specified pool in each instance of the story. For example, consider the state order_food:

[subject] ordered a plate of [dessert]

In a specific instance of the state, the roles subject and dessert would be occupied by randomly chosen fillers, such as Alice and chocolate.

In Fig. 1 and Table 1, we show the specific schema used in our experiments.

Figure 1 Story graph for role-filler binding experiments.

Each edge indicates a possible transition. For states with multiple outgoing transitions, each outgoing transition is equally likely.

Table 1 Story states for role-filler binding experiments.

We provide the text of each state of the story, where the bracketed roles are substituted by specific fillers in each story.

BEGIN	begin [Subject]	
ORDER DRINK	order_drink [Subject] [Drink]	
EXPENSIVE	too_expensive [Subject]	
SIT	sit_down [Subject] [Friend]	
INTRO	emcee_intro [Emcee] [Poet]	
POETRY	poet_performs [Poet]	
DECLINE	subject_declines [Subject]	
PERFORM	subject_performs [Subject] [Friend]	
GOODBYE	say_goodbye [Subject] [Friend]	
ORDER DESSERT	order_dessert [Subject] [Dessert]	
END	end [Subject]	

Main task

Our main task tests each model’s ability to extract role-filler pairs based on the structure of an input sentence. We presented networks with stories generated from the specified schema, and then queried the networks for the filler corresponding to a specified role. For instance, the network might receive the input

begin alice sit alice bob poet_performs chris subject_performs alice bob say_goodbye alice bob end alice qpoet

In this case the correct output is chris, because chris is the filler corresponding to the role poet in the schema we define in Fig. 1 and Table 1.

We train each model on a certain pool of fillers, and construct test sets in which inputs contain fillers not seen during training. The position of the filler corresponding to a given role is not necessarily the same in each story because transitions between states are probabilistic. To successfully complete these tasks, the models must learn to extract the filler corresponding to each role, store these role-filler pairs during the input sequence, and select the correct filler to output after receiving the query.

We note that our goal is to identify models that can perform role-filler binding in this setting, rather than to prove that some specific network architecture performs best on a downstream language processing task. Therefore our evaluation metrics are meant to show that networks are capable of learning the task, and we do not compare hyperparameters or sample efficiency in this work.

Input representations

We represented each word of the input sentence as a vector. This vector is a fixed mapping from a one-hot V-dimensional vector that encodes word identity, to a randomly generated 50-dimensional word embedding. Each 50-dimensional filler vector that is not seen during training represents a novel filler, since it constitutes a unit in the V-dimensional word-identity space that is not used during training.

In our first three experiments, each index of the vector is independently drawn from a N(0,1) distribution, and then the vectors are normalized to have unit Euclidean norm. We tested different distributions in our tests of correlation violation and retention, which we describe in the “Correlation Violation and Retention Tests” section of our Methods. We used randomly generated vectors rather than pretrained word vectors such as word2vec (Mikolov et al., 2013) to show that networks can learn role-filler binding even without strong prior information about semantic similarities between input words.

We sequentially fed the words of the story into the networks, followed by a query word indicating which filler to retrieve. The network then outputted a 50-dimensional vector, and we computed the cosine similarity between this prediction and each vector in the experiment’s corpus, choosing the most similar word as the network’s prediction.

We designated one word in the vocabulary as a padding word and inserted it into a randomly chosen location in the input story, to force the network to learn representations of the schema that are robust to small position shifts.

To ensure that all inputs have the same number of words, we appended the padding word between the end of a story and the appearance of the query.

Training regimes

We ran experiments with two types of training regimes: Limited Filler Training and Unlimited Filler Training. In “Limited Filler Training” experiments, we substituted roles with fillers drawn from a small, finite pool of fillers (with six possible fillers for each role). Within this category of experiments, we tested the network using examples with previously seen and previously unseen fillers. When testing with previously seen fillers, the pool of fillers is the same during training and testing. When testing with previously unseen fillers, we drew from disjoint pools of fillers during training and testing, meaning that the network needed to perform role-filler binding with fillers it had never seen before. In all experiments, we ensured that the train and test set contained distinct input sequences (i.e., they could not contain inputs with both the same sequence of states and the same role-filler pairs).

In “Unlimited Filler Training” experiments, we randomly generated a new vector for each filler in each input story during both training and testing, rather than using a finite pool of fillers. In this case, during both training and testing, the network was continuously asked to perform role-filler binding using previously unseen fillers.

Prediction method

To determine the network’s prediction, we used networks in which the final layer has 50 nodes. We computed the cosine similarity between the output vector and the vector embedding of each word in the experiment’s corpus, and selected the word with the highest cosine similarity to the network’s output vector.

The set of possible words is the corpus created by combining the words seen in all stories in a particular training batch. For fixed embeddings, the corpus therefore consists of all of the words that occur in the stories generated for a particular experiment. For experiments in which we generated a new random embedding for each story, the corpus also includes all the filler vectors newly generated for stories in that particular batch.

Chance rates

In each experiment, the network’s chance rate depends on the number of words in the corpus.

In Experiments 1 (Limited Filler Training, tested with previously seen fillers) and 2 (Limited Filler Training, tested with previously unseen fillers), the network must choose from a corpus of 50 words, corresponding to a chance rate of 2%.

In Experiment 3 (Unlimited Filler Training, tested with previously unseen fillers) the network must choose from all the words in the initial story corpus (30) and the newly generated representations for each story in the batch. Since we used a validation batch size of 16, and six new filler vectors are generated for each input, this results in a total of 30 + 16 × 6 = 126 words, for a chance rate of 0.8%.

Epoch sizes

In Experiment 1 we used 63,175 train and 15,794 test stories. We created these stories by first generating 5,000 stories for each of the 24 possible state sequences, with six choices for each type of role, and then removing repeated story sequences. 80% of stories were used for training and the remaining 20% for testing. In Experiment 2 we trained networks using the same set of train stories, and tested networks with a separate set of 3,658 test stories. We created this separate set of test stories by first generating 250 stories for each of the 24 possible state sequences, using a pool of fillers that did not overlap with fillers used in the train set, and then removing repeated story sequences.

In Experiment 3 we used 112 train and 112 test story frames. The number of story frames corresponds to the number of possible queries (queries that can be answered using the information in the story; for instance, some stories may not include an Emcee and therefore the input must not use QEmcee as a task) for each possible traversal through the story graph. This gives us 112 story frames, which are filled with newly generated fillers in each example.

Models

We tested four recurrent neural network (RNN) architectures. RNNs are a class of neural network architectures with weights that form directed cycles. The cycles form feedback loops that allow networks to maintain an internal state. The structure of RNNs allows us to provide the input story one word at a time, followed by the query. The feedback loops allow for a form of short-term memory (where we define “short-term” as the timescale of a single story), with which they could maintain relevant parts of the story. We tested multiple RNN network architectures, to investigate which memory components (if any) are sufficient for role-filler binding. For each of our architectures we used 50 hidden units and a learning rate of 1e−4.

In addition to a standard RNN, we tested Long Short-Term Memory (LSTM), Fast Weights, and Differentiable Neural Computer (DNC) architectures. We used layer normalization for the RNN, LSTM and Fast Weights architectures. Layer normalization re-centers and re-scales the networks’ layers and serves to stabilize the network dynamics (Ba, Kiros & Hinton, 2016).

The LSTM consists of an RNN with gates to control what the internal state stores, forgets, and displays to the rest of the network (Hochreiter & Schmidhuber, 1997). The Fast Weights architecture consists of an RNN with a matrix of quickly changing “fast weights” (Ba et al., 2016). This extra matrix of weights allows for auto-associative memory, and the combination of the quickly changing fast weights matrix and more slowly changing standard weights is inspired by different speeds of change in biological neuronal connections (Martin, Grimwood & Morris, 2000). The DNC is an RNN with an LSTM “controller” that learns to read from and write to an external buffer (Graves et al., 2016). The network can use an external buffer as a “mental scratchpad” to store and retrieve memories. The controller must learn how to use this external buffer. The combination of a controller and external memory buffer is inspired by interactions between the hippocampus and cortex in the human brain. This interaction plays a key role in human memory (O’Reilly et al., 2014). Our DNC model has a memory size of 128, a word size of 20, 1 write head, and 1 read head. These networks have shown success on a range of tasks including speech recognition (Graves, Mohamed & Hinton, 2013), language modeling (Mikolov & Zweig, 2012) and associative recall (Graves, Wayne & Danihelka, 2014).

Decoding analysis

We performed decoding analyses to analyze the mechanisms underlying task performance.

For each model, we recorded network activity after the model received each word in an input sequence. For each example and each model, this resulted in one 50-dimensional vector of hidden unit activity per input word. The Fast Weights and DNC networks consist of two memory components (the hidden state and an external memory component), and we recorded the values of the hidden states and the external memory components. From the Fast Weights network, we obtained a 50-dimensional vector of hidden unit activity, and a 50-by-50-dimensional matrix (which we flatten into a 2,500-dimensional vector) of fast weights activity, after each word in the input sentence. From the DNC, we obtained a 50-dimensional vector of hidden unit activity, and a 128-by-20-dimensional matrix (corresponding to 128 memory slots each of size 20, which we flatten into a 2,560-dimensional vector) of memory buffer activity, after each word in the input sentence.

We constructed 100 input sequences with the same story frame and completed each sequence with distinct fillers (i.e., the frame text for each sequence is identical, but the fillers were different). For each role, we trained a ridge regression mapping (with regularization strength 1.0) between each memory state vector and correct output filler, using 80 of the sequences for training. For each role, this resulted in six regression mappings: four mappings from each model’s 50-dimensional hidden state to the 50-dimensional correct output filler, one mapping from the 2,500-dimensional vector of fast weights to the 50-dimensional correct output filler, and one mapping from the 2,560-dimensional DNC memory buffer to the 50-dimensional correct output filler. Then on each of the remaining 20 sequences, we used this mapping to predict the output filler. We ranked each corpus vector in terms of its cosine similarity with the predicted output, and computed the ranking score (1−rankofactualoutputcorpussize) for each test sequence. These ranking scores have a maximum score of 1, with a chance rate of 0.5.

Correlation violation and retention tests

Previous work suggested that connectionist networks learn statistical correlations rather than relational structure; for example, a recent study showed that network performance falls below chance when the network receives test examples with different statistical structure from training examples (Puebla, Martin & Doumas, 2019). We constructed two experiments to assess how well the models generalize to role-filler bindings that violate correlations seen during training.

The first task probed whether networks can perform role-filler binding on test examples that break correlations observed during training. Networks were trained on stories constructed from the frame begin subject sit subject friend announce emcee perform poet consume dessert drink goodbye. In this experiment, the fillers for subject, friend, emcee, and poet were drawn from a pool of fillers, while all other words stayed constant in each example. There were 1,000 possible fillers, and during training each role could be filled by a fixed subset of 750 of these fillers (i.e., each filler is excluded from one of these four roles during training). During test, each role is filled only by fillers that were excluded from that role during training, to test whether networks learn a strategy for role-filler binding that is robust to violations of correlations seen during training.

Our second test is motivated by humans’ ability to generalize role-filler binding to arbitrary fillers that violate correlations seen in previous experiences, while also retaining knowledge of the correlations seen during training. Continuing our example from the beginning of this section, if during training we usually see Alice as the customer and Bob as the barista, we can remember this statistic while still extracting the associations barista:Alice, customer:Bob from the sentence Bob ordered a tea from Alice.

In this experiment, we consider three distributions, each of which starts from a 50-dimensional vector drawn from a N(0,1) distribution. In the first distribution, which we call distribution X, with 90% probability we add 0.5 to each even index, and with 10% probability we subtract 0.5 from each even index. In the second distribution, which we call distribution Y, with 10% probability we add 0.5 to each even index and with 90% probability we subtract 0.5 from each even index. In the third distribution, which we call distribution Z, we perform no extra additions or subtractions to any of the indices. Each vector is normalized to have Euclidean norm 1.

During training and testing, each frame word (i.e., each non-filler word) is drawn from distribution Z. During training, three of the roles (Dessert, Emcee, Poet) are drawn from distribution X while the other three roles (Subject, Friend, Drink) are drawn from distribution Y.

To test networks’ ability to perform role-filler binding on arbitrary fillers, we test networks in four settings: each filler is drawn from distribution X, each filler is drawn from distribution Y, each filler is drawn from distribution X or Y with equal probability, or each filler is drawn from distribution Z.

To test networks’ ability to retain statistical correlations, we test networks in two settings: In the first setting, we construct ambiguous input sentences that contain insufficient information for determining the correct response, following St. John & McClelland (1990). In these ambiguous experiments, the queried filler is substituted with a vector consisting of all zeros. This vector is not present in the corpus and has no bias towards greater even or odd indices. We compare the even and odd indices of the 50-dimensional vector predicted by the network, to test whether the networks’ responses to ambiguous examples mirror regularities seen during training.

In the second setting, we test the networks on fillers that are drawn from distribution Z, and simultaneously measure accuracy of recall and statistical learning (i.e., whether the network shows biases mirroring the statistical regularities seen during training). Accuracy of recall was operationalized as whether the networks produce a vector vpred that is closer (in terms of cosine similarity) to the correct answer vcorrect than to other vocabulary items. To measure statistical bias, we compared the even and odd indices of vpred − vcorrect to test whether the predictions are biased towards statistics seen during training even when they provide the correct answer.

We also performed an experiment in which the structure of the input story was drastically different from the structure seen during training. Networks were trained on the story begin subject sit subject friend announce emcee perform poet consume dessert drink goodbye, and tested on a shuffled version of the story: consume dessert drink goodbye begin subject sit subject friend announce emcee perform poet.

Code

We used TensorFlow to implement our experiments, adapting existing architecture implementations from Mohandas (2018) and DeepMind (2018). We used Coffee Shop World to generate the stories used in this experiment. This generator is available on GitHub (https://github.com/PrincetonCompMemLab/narrative).

The code used to generate data, run experiments, and generate the plots in this article is available on GitHub at https://github.com/cchen23/generalized_schema_learning/. We also include pre-generated data and checkpoints of trained networks.

Results

Experiment 1: limited filler training, tested with previously seen fillers

In the Limited Filler Training experiment, train and test inputs contain fillers drawn from the same pool of fillers. During test, the networks are provided with new stories—they had seen each word of each input sequence during training, but never with this particular permutation of words. This experiment tests whether networks possess the ability to learn associations between roles and fillers, given stories generated from an underlying schema.

As we show in Fig. 2, each architecture performs the experiment task at a significantly above chance rate. While the basic RNN learns more slowly than other networks, its architecture is sufficient to learn and apply a schema to situations in which it has seen the fillers before, in a slightly different context.

Figure 2 Test scores for Experiment 1.

Each architecture is able to learn to perform role-filler binding on a story it has not previously seen, if it has encountered each of the story’s words during training. The chance accuracy rate is 2%, bar heights denote mean training accuracies, and error bars denote maximum and minimum accuracies over three trials. Full learning curves are available in the Supplemental Material.

Experiment 2: limited filler training, tested with previously unseen fillers

We conducted a second Limited Filler Training experiment, in which the networks were tested on stories containing fillers they had not encountered during training. This experiment tested whether networks can learn to not only perform role-filler binding, but also to generalize their schematic knowledge to fillers they have never encountered.

All networks fail to do so. While all networks perform far above chance on previously seen stories presented during training, the test accuracy of each network remains at 0% (the chance accuracy rate from guessing random vectors is 2%). The test accuracy lies below the chance rate because the networks overfit to the specific fillers seen during training. When we examined the specific words predicted by the networks, we found that the networks always predict fillers from the train set. In this experiment the train and test fillers are drawn from disjoint pools. Therefore the networks always predict the wrong response during test.

Experiment 3: unlimited filler training, tested with previously unseen fillers

We conducted an Unlimited Filler Training experiment, in which we tested the networks with previously unseen fillers. We constructed train and test sets in which fillers were represented by new randomly generated fillers in each example; thus, the networks need to generalize to previously unseen fillers to succeed in both the train and test sets.

As we show in Fig. 3A, all architectures reach above-chance test accuracy, showing that all architectures are sufficient for some amount of generalization to previously unseen fillers, when given a train set with an unlimited pool of fillers.

Figure 3 Overall and query-split accuracies for Experiment 3.

(A) Overall accuracies. Three architectures reach above-chance test accuracy in all three trials, showing that certain networks perform some amount of generalization when forced to do so during training. (B) Query-split accuracies. The LSTM and RNN learn to generalize only on the QSubject task. The DNC and Fast Weights networks learn to solve all six tasks. The figure legend (C) indicates the color that corresponds to each query. The chance rate is 0.8%, bar heights denote mean accuracies, and error bars denote maximum and minimum accuracies over three trials. Full learning curves are available in the Supplemental Material.

The different degrees of success shown in Fig. 3A reflect the uneven difficulty of queries. In Fig. 3B, we show network performance separated by query. The RNN does not consistently succeed in answering any role query, and the LSTM succeeds only in answering queries for the Subject role. The Fast Weights and DNC networks learn to perform role-filler binding for all queries. As indicated by the schema structure in Fig. 1, the filler corresponding to the Subject role is easiest to identify, as it always occurs at the same location in the story. In contrast, all other roles have variable locations within the story, depending on the stochastically chosen sequence of story states. We experimented with larger LSTM networks that are either wider (with 2,500 hidden units) or deeper (with three layers). These networks reach test accuracies of between 0.25 and 0.30. This shows that the gap in performance between the LSTM model and the (better-performing) Fast Weights and DNC models can not be closed simply by performing these expansions to the LSTM.

The results of this experiment show that if networks receive an unlimited pool of train fillers, then some networks learn to perform role-filler binding, while simpler networks either fail to learn role-filler binding, or learn to perform role-filler binding for only the simplest queries.1

Decoding analysis

We performed decoding analyses on the four networks trained in Experiment 3, to gain insight into how and if the memory components aid in learning role-filler binding. We find that the ability to decode correct fillers, at the time that the network is presented with the query, corresponds to networks’ success in role-filler binding. We show the ranking scores (averaged over three trials) in Fig. 4. We include the ranking scores, separated by trial, in the Supplemental Materials.

Figure 4 Decoding scores for Experiment 3.

For the RNN (A) and LSTM (B), which are unable to solve any of the tasks other than QSubject, decoding scores of each of the fillers other than the Subject are around the chance rate at the end of the input sequence. The Fast Weights and DNC architectures show a similar trend in the hidden internal state of the controllers: decoding scores of the hidden states peak when the networks receive the respective filler in the input sequence, then decline as the network receives more words (C and D). In comparison, the decoding scores of the external memory components (the Fast Weights matrix and the DNC’s external memory buffer) increase when the network receives the corresponding filler in its input and the scores remain high throughout the input sequence (C and D). The figure legend (E) indicates the line representing the decoding score for each filler. The chance rate is 50%. These decoding scores are averaged over three trials. Decoding scores separated by trial are available in the Supplemental Materials.

From the RNN’s and LSTM’s hidden states we can decode only the Subject at an above-chance rate at query-time, mirroring these network’s ability to only solve QSubject tasks (Figs. 4A and 4B).

With the Fast Weights architecture, we performed decoding using either the controller’s hidden state or the set of associative fast weights. We show these decoding scores in Fig. 4C. The decoding scores of the controller’s hidden state mirror those of the LSTM network’s hidden state: The scores peak when the network receives the filler in its input, then decline as the network receives more words. (An exception is decoding scores for the Subject filler. This could be due to the fixed location of the Subject filler, and the non-fixed locations of the other fillers.) We see this trend regardless of whether the network successfully retrieves a certain type of filler. In contrast, the decoding scores of the Fast Weights matrix increase when the network receives the corresponding filler in its input and remain above chance at query-time.

We see a similar pattern between standard and external memory components with the DNC (Fig. 4D), where the decoding scores of the DNC’s controller’s hidden state mirror those of the LSTM, and the decoding scores of the DNC’s external memory matrix mirror those of the Fast Weights matrix. These results suggest that networks learn to solve tasks by storing the relevant information using their external memory components (either the external memory buffer or the fast weights matrix), while the controller acts as a conduit to receive these words and move them to the external memory component.

Furthermore, the read and write weights of the DNC indicate that the network learns to store and retrieve role-filler bindings using a location-based strategy. The read weights influence where in the external memory buffer the network should read from, and the write weights influence where in the external memory buffer the network should write to. In Fig. 5 we show the maximum write weights at input timesteps corresponding to the appearance of fillers corresponding to each role, and the maximum read weights at the timestep at which the network makes its prediction. The rows showing network write weights show that the network associates different slots in memory with each role. Furthermore, these results show that the distribution of network read weights when a query word occurs in the input sequence, corresponds to the distribution of network write weights when the corresponding filler occurs in the input sequence. This suggests that the network saves and retrieves role-filler pairs using a location-based strategy.

Figure 5 Maximum read and write weights for DNC external memory buffer.

Read and write weights indicate that DNC networks use a location-based strategy. For all examples in the test set of Experiment 3, we determined the index of the external memory buffer with the highest read or write weights at each input timestep. The network associates different external memory slots more strongly with each role. Correspondence between write weights when the network receives the filler for role “R”, and read weights when the network is queried for role “R” indicate that networks use a location-based strategy to save and retrieve role-filler pairs. (A) Histograms of maximum write weights, for input timesteps in which the network receives the filler corresponding to the role denoted by y-axis labels. (B) Histograms of maximum read weights, for each input timestep in which the network receives the query denoted by y-axis labels. (C) Pearson correlations between the distribution of maximum write indices when the network receives fillers corresponding to each role (on the x-axis), and the distribution of maximum read indices when the network receives queries asking for each role (on the y-axis). This figure corresponds to one of three trials. Analyses for two other trials are available in the Supplemental Materials.

These findings suggest that networks that learn to perform role-filler binding also learn to store relevant information in external memory components.

Correlation violation and retention tests

In our test of correlation violation, we trained networks using stories with strong role-filler correlations. We constructed these correlations by excluding each filler from a specific role during training. We then constructed a test set that deliberately breaks the correlations seen during training. In the test set, we associated each role only with fillers that were excluded from that role during training. When networks receive inputs with role-filler bindings that violate correlations seen during training, they accurately extract the correlation-violating role-filler bindings. All networks reach above chance accuracy on this test set (the DNC and Fast Weights architectures reach 100% accuracy), as we show in Fig. 6, showing that the networks can generalize to role-filler bindings that violate statistics seen during training.

Figure 6 Test accuracy in correlation violation experiment.

Some networks learn to perform role-filler binding, even when test examples break role-filler correlation observed during training. The chance accuracy rate is 0.02%, bar heights denote mean accuracies, and error bars denote maximum and minimum accuracies over three trials.

In our test of correlation retention we trained the Fast Weights and DNC models on examples in which the even indices of filler vectors were drawn from either N(0.5,1) or N(−0.5,1), while odd indices of filler vectors (and all indices of non-filler vectors) were drawn from N(0,1). Fillers for three arbitrarily selected roles (Dessert, Emcee, Poet) were more likely to contain higher even indices (distribution X), while fillers for the other three roles (Subject, Friend, Drink) were more likely to contain lower even indices (distribution Y).

We tested networks on examples in which the fillers are each drawn from distribution X, each from distribution Y, each from distribution X or Y with equal probability, or each from distribution Z. As we show in Figs. 7 and 8, the Fast Weights and DNC networks both reach above-chance accuracy for each of these distributions. We note that, although the networks see non-filler words drawn from distribution Z during training, all filler words presented during training are drawn from distributions X or Y.

Figure 7 Test accuracy in correlation retention experiment: Fast Weights.

The Fast Weights model reaches high accuracies when extracting role-filler pairs from examples in which each filler is drawn from distribution X (A), each from distribution Y (B), each from distribution Z (C), or each from distribution X or Y with equal probability (D). Bar heights denote mean test accuracy, and error bars denote minimum and maximum accuracy across five trials. For instances in which all five trials achieved the same accuracy, no error bars are shown.

Figure 8 Test accuracy in correlation retention experiment: DNC.

The DNC model reaches high accuracies when extracting role-filler pairs in inputs in which each filler is drawn from distribution X (A), each from distribution Y (B), each from distribution Z (C), or each from distribution X or Y with equal probability (D). Bar heights denote mean test accuracy, and error bars denote minimum and maximum accuracy across five trials. For instances in which all five trials achieved the same accuracy, no error bars are shown.

We then tested whether networks retain information about the distributions associated with each role during training. We constructed test examples in which each filler is drawn from distribution X or Y with equal probability, and then the queried filler vector is replaced with a vector of all zeros. Distributions X and Y differ in the relative values of the even indices of the word vectors. In Figs. 9 and 10 we show that the Fast Weights predictions mirror these biases. Figure 9 shows that the distributions of the even and odd indices of vectors predicted in response to each query (the predictions were aggregated over five trials) are biased towards the training distributions for each role. To measure the extent of these biases, we computed the difference between the predicted even and odd indices, using the t-statistic between the predicted even and odd indices in response to each filler. We show the t-statistic for each of five trials in Fig. 10. The Fast Weights network reliably demonstrates biases towards the training distributions. The DNC does not display these biases in response to our probes.

Figure 9 Histograms of predicted even and odd indices.

We show histograms of even and odd indices of predicted filler vectors, in response to examples in which (at test) the queried role is substituted with a vector of all zeros, and all other fillers are drawn from distribution X (higher even indices on average) or distribution Y (lower even indices on average) with equal probability. During training, filler vectors corresponding to three roles (Dessert, Emcee, Poet, shown in orange) tend to have greater values in even indices of the vector, while filler vectors corresponding to the other three roles (Subject, Friend, Drink, shown in blue) tend to have lower values in even indices of the vector. The Fast Weights networks’ predictions display a bias towards the distributions seen during training (A), while the DNC networks do not display this bias (B). Results are aggregated across five trials.

Figure 10 t-Statistics of predicted even and odd indices: ambiguous queries.

We show t-statistics comparing even and odd indices of predicted filler vectors in response to examples in which the queried role is substituted with a vector of all zeros, and all other fillers are drawn from distribution X (higher even indices on average) or Y (lower even indices on average) with equal probability. During training, filler vectors corresponding to three roles (Dessert, Emcee, Poet, shown in orange) tend to have greater values in even indices of the vector, while filler vectors corresponding to the other three roles (Subject, Friend, Drink, shown in blue) tend to have lower values in even indices of the vector. These t-statistics show that the Fast Weights networks’ predictions reflect role statistics seen during training (A). The DNC networks do not display this bias (B).

In non-ambiguous examples, networks exhibit these biases as well. To correctly answer a query, networks need to predict a vector that is closer to the queried filler than to any other corpus word. Importantly, these predictions are robust to small differences from the actual filler (so long as the predictions remain closer to the queried filler than to any other word in the corpus)—this provides the network with a means of showing subtle statistical biases even when the response is correct. We compared the difference between the predicted and actual fillers, and in Fig. 11 we show that the Fast Weights network retains statistical correlations seen in training while predicting the correct response to arbitrary role-filler pairs.

Figure 11 t-Statistics of predicted even and odd indices: non-ambiguous queries.

We show t-statistics comparing the distance from even and odd indices of predicted filler vectors to the correct filler values, in examples in which each filler is drawn from distribution Z (each index of the filler vector is drawn from N(0,1)). During training, filler vectors corresponding to three roles (Dessert, Emcee, Poet, shown in orange) tend to have greater values in even indices of the vector, while filler vectors corresponding to the other three roles (Subject, Friend, Drink, shown in blue) tend to have lower values in even indices of the vector. These t-statistics show that Fast Weights networks can recall role statistics seen during training (A), and also produce the correct answer in almost all test instances (see Figs. 7 and 8). The DNC networks do not display this bias (B).

In summary: In these tests of correlation retention, the Fast Weights networks generalize to examples that violate correlations seen during training, and also display some knowledge of correlation statistics seen during training.

No networks consistently succeed in performing role-filler binding with shuffled story test sets.

Discussion

In previous work, models successfully perform role-filler binding when they are given explicitly labeled inputs (Kriete et al., 2013; Elman & McRae, 2019; Franklin et al., 2019), or when they are given train and test examples with the same set of fillers (St. John & McClelland, 1990; Miikkulainen & Dyer, 1991; Hinaut & Dominey, 2013). They fail when given test stories that include previously unseen fillers that violate correlations seen during training (Puebla, Martin & Doumas, 2019). In our experiments, we find that some models can perform role-filler binding on inputs that are not explicitly labeled with role-filler pairs. Moreover, they can do so on novel fillers and on bindings that violate correlations seen during training, while retaining knowledge of statistical correlations observed during training.

In our experiments we found that networks trained on stories with a small pool of train fillers (as in Puebla, Martin & Doumas (2019)) fail at generalizing to novel fillers, while the networks that both have external memory and are trained on a larger pool of fillers succeed. This suggests that networks must see a sufficiently diverse set of fillers during training, in order to learn to separate the representations of roles and fillers. Furthermore, the successful networks contain external memory (the set of fast weights or the DNC’s external memory buffer). Our findings suggest that external memory could be an important architectural component for networks to learn role-filler binding. Previous work emphasized the importance of having independent bindings (i.e., where the binding relations are represented separately from the bound entities), and of having dynamic bindings (i.e., where the same filler can be bound to different roles)—according to these definitions, encoding the role-filler binding by storing a modified representation of the filler fails to preserve independence between representations of roles and fillers (Fodor & Pylyshyn, 1988; Hummel et al., 2004).

Networks can come closer to clean combinatorial generalization by forming memory representations that are separate from the learnable network weights. We hypothesize that the Fast Weights and DNC models’ external memory components allow them to preserve this independence, and therefore generalize farther from the statistics of the training environment. The availability of an external memory buffer makes it possible to store role-filler bindings by placing the filler in a certain location in memory. When a role is queried, the model can extract a filler from the corresponding location in memory, without necessarily relying on characteristics of the filler vector itself. This means that it can perform this task for any arbitrary filler vector.

Furthermore, despite the trade-off between sensitivity to training distributions and generalization to arbitrary distributions, Fast Weights networks retain information about statistical regularities seen during training. The networks display subtle biases towards training distributions while remaining closest to the correct filler vector. This is possible because output words are selected as the word in each batch’s corpus that is closest to the network’s predictions.

Our experiments include bindings that violate correlations seen during training, and fillers that are not seen during training. This means that, in order to successfully perform our tasks at test time, networks must store role-filler pairs in a way that is not specific to role-filler correlations seen during training. Furthermore, our experiments used randomly generated vectors (rather than pretrained word vectors) to represent input words, meaning that networks could not rely on pre-encoded linguistic knowledge.

Our decoding analyses suggest that the successful networks learn to store fillers in their external memory. Specifically, we were able to decode fillers at query-time from the Fast Weights and DNC models’ external memory components (but not from the controller); this suggests that these networks perform role-filler binding by first storing bindings in the external memory components, and then retrieving the correct binding upon receiving the query. The read and write weight distributions of the DNC suggest that the network learns to use certain locations in memory to store fillers corresponding to each role. External memory provides a separate storage space, which gives the model a way to encode a filler’s role by storing it in a certain location in external memory, without needing to encode role-filler bindings in a modified representation of the filler.

While the Fast Weights model can perform role-filler binding on arbitrary fillers, we also found that its responses are biased towards statistical regularities seen during training. Because correct responding was operationalized using a “nearest neighbor” metric that is robust (to some degree) to distortions in the response vector, Fast Weights networks can manifest subtle biases towards statistical regularities from the training phase without compromising (nearest-neighbor) accuracy. The DNC model performs role-filler binding on arbitrary fillers without displaying biases towards statistical regularities seen during training. This suggests that the DNC model learns a procedure for role-filler binding that stores filler vectors without imposing a bias towards a specific distribution. This could be due to the presence of read and write gates (the DNC model contains gates which control external memory updates after each stimulus word, while the Fast Weights models performs a memory update after each word) or distinct memory slots (the DNC model contains explicit memory slot locations and addressing mechanisms, while the Fast Weights model updates an external memory matrix using the outer product of the network’s hidden state). Future theoretical and empirical work could further investigate these differences.

We note that our results show that some models are sufficient for performing role-filler binding tasks, but we do not conclude that all standard RNN and LSTM models are unable to solve these tasks. In our experiments, larger LSTM models (with 2,500 hidden units rather than 50, or with three layers rather than one) did not solve all tasks. However, future work with even larger LSTM models might be able to identify LSTMs that can learn role-filler binding without explicit external memory.

Networks fail when given stories presented in a radically different order, showing that they rely on structural similarities between train and test sentences to identify and store role-filler pairs. Previous work showed that sufficiently large neural networks can learn arbitrary functions of the training data if they are trained on randomly assigned labels. This work used a neural network to memorize a function mapping from a set of images to randomly assigned labels (Zhang et al., 2017)—by contrast, in our shuffling experiments, networks are trained on a fixed order of inputs before being tested on a different (shuffled) order. We speculate that, if a sufficiently large network is trained on sufficiently many examples with shuffled stages, it could learn a function that generalizes to shuffled test steps. It could do this by learning to identify a stage based on the frame-words of that stage only, while ignoring the sequence of the story’s stages. Future work could explore training regimes that use shuffled stories during training, to see whether this could make models robust to test stories with shuffled segments. Future work could also test how the loss of certain architectural components affects the ability to perform schema-learning, by lesioning certain components of an artificial network. Our experiments indicate that the diversity of training examples influences whether networks learn to perform role-filler binding. Future work could investigate how much a pool of train fillers needs to be expanded to allow networks to learn a representation that extends to previously unseen fillers. Furthermore, the psychology literature indicates that schemata can encode biases and stereotypes, affecting how humans interpret new information and recall previous information (Frederic Charles Bartlett, 1932). Future work could explore whether and how models adapt to changes in the underlying schema over the course of training, how examples provided during training might translate into biases encoded in schemata, and how these principles apply in situations where multiple schemata need to be learned (Franklin et al., 2019). The impact of external memory components in our experiments suggests that episodic memory (a cognitive ability that corresponds to the external memory components) may play an important role in applying schemata to novel fillers, and further work could investigate this connection.

Another important future direction is to test models of role-filler binding against neural data. For example, Martin & Doumas (2017) and Martin (2020) recently argued that models that use the timing of neural firing to bind fillers to roles (Doumas, Hummel & Sandhofer, 2008) are supported by data showing neural entrainment during language processing, at frequencies corresponding to multiple layers of the representational hierarchy (Ding et al., 2017; but see Frank & Christiansen, 2018). Analogously, in future work, one could use neural data to test key predictions of our neural network models with external memory. A challenge in this respect is that information stored in external memory is thought to be latent or activity-silent (i.e., stored in network weights and not in active patterns of neural firing) and thus should not be decodable from neural activity measurements. One way to address this challenge would be to take advantage of newly-developed paradigms for probing the contents of activity-silent memory (e.g., using transcranial magnetic stimulation to evoke the information into an actively-represented state) (Rose et al., 2016).

Conclusions

Our experiments find networks that can perform role-filler binding, where we define role-filler binding as the ability to bind arbitrary fillers to certain roles without receiving explicitly labeled bindings, even when these pairings violate correlations seen during training. Successful models can generalize to previously unseen fillers when given external memory and when given novel fillers during training. These models can perform role-filler binding with novel fillers, and with fillers that violate role-filler correlations seen during training. This suggests that the networks learn to extract relations from the structure of the input, rather than relying on role-filler correlations seen during training.

Previous work has suggested that indirection can support variable bindings (Kriete et al., 2013), and Complementary Learning Systems theory hypothesizes that the coordination of two distinct systems allows models to both learn overall structures shared between experiences and rapidly learn new items (McClelland, McNaughton & O’Reilly, 1995). While future work could explore whether networks without external memory can also perform these tasks, our experiments show that, at the very least, there are models with external memory that learn role-filler binding. These findings provide a possible mechanism for connectionist architectures to learn role-filler binding relations, which are a central component for learning flexible, structured cognitive representations.

Supplemental Information

Supplemental Information 1 Test accuracy for Experiment 1. (Limited Filler Training Tested with Previously Seen Fillers, originally shown in Figure 2.) The chance accuracy rate is 2.3%.

Click here for additional data file.

Supplemental Information 2 Train and test accuracy for Experiment 2. (Limited Filler Training Tested with Previously Unseen Fillers, originally shown in Figure 3.) The chance accuracy rate is 2.3%.

Click here for additional data file.

Supplemental Information 3 Maximum Read and Write Weights for DNC External Memory Buffer, Trial 2. (Weights for Trial 1 is shown in Figure 5.).

Click here for additional data file.

Supplemental Information 4 Maximum Read and Write Weights for DNC External Memory Buffer, Trial 4. (Weights for Trial 1 is shown in Figure 5.).

Click here for additional data file.

Supplemental Information 5 Decoding Scores for Experiment 3, Trial 1. (Decoding scores aver- aged across all three trials are shown in Figure 4.).

Click here for additional data file.

Supplemental Information 6 Decoding Scores for Experiment 3, Trial 2. (Decoding scores aver- aged across all three trials are shown in Figure 4.).

Click here for additional data file.

Supplemental Information 7 Decoding Scores for Experiment 3, Trial 3. (Decoding scores aver- aged across all three trials are shown in Figure 4.).

Click here for additional data file.

Any opinions, findings, and conclusions or recommendations expressed in this material are those of the author(s) and do not necessarily reflect the views of the Office of Naval Research or the U.S. Department of Defense.

Additional Information and Declarations

Competing Interests

Author Contributions

Data Availability

1 The focus of our experiments here is on qualitatively comparing different network architectures; while differences in properties of the individual simulations (e.g., vocabulary size) might affect the exact level of performance that is reached, the qualitative patterns reported here can not be explained in terms of simple factors (e.g., larger vocabulary leading to worse performance). For example, our results show that networks fail to correctly identify the filler in Experiment 2 (an experiment with a smaller vocabulary), while succeeding on the task when given unlimited train fillers in Experiment 3 (an experiment with a larger vocabulary).

The authors declare that they have no competing interests.

Catherine Chen conceived and designed the experiments, performed the experiments, analyzed the data, prepared figures and/or tables, authored or reviewed drafts of the paper, and approved the final draft.

Qihong Lu conceived and designed the experiments, authored or reviewed drafts of the paper, and approved the final draft.

Andre Beukers conceived and designed the experiments, authored or reviewed drafts of the paper, and approved the final draft.

Christopher Baldassano conceived and designed the experiments, authored or reviewed drafts of the paper, and approved the final draft.

Kenneth A. Norman conceived and designed the experiments, authored or reviewed drafts of the paper, and approved the final draft.

The following information was supplied regarding data availability:

Code and data are available at GitHub: https://github.com/cchen23/generalized_schema_learning.

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
