# Peer review of "Learning to perform role-filler binding with schematic knowledge"

_PeerJ, doi:10.7717/peerj.11046_

## Round 0.1 · original submission · Major Revisions

Your manuscript has now been seen by 3 reviewers. You will see from their comments below that while they find your work of interest, some major points are raised. We are interested in the possibility of publishing your study, but would like to consider your response to these concerns in the form of a revised manuscript before we make a final decision on publication. We therefore invite you to revise and resubmit your manuscript, taking into account the points raised. Please highlight all changes in the manuscript text file.

Reviewer 1 ·

Basic reporting

The abstract does not make it as clear as it could be what a network understands in this context, e.g., what is a structural relationship, what role-filler binding is, and what schemas are.

In the intro I would have expected some overview of the history into schema theory, and especially into the models proposed by other scientists. Schemas presumably have a long history in psychology and there are a multitude of theories and models out there to deal with predicting and understanding the mechanisms and functions involved. This seems pivotal to making a case for why this manuscript is valuable as well as orienting the reader.

The paragraph starting at line 37 is lacking. What is the difference between the work here and prior work? On lines 45-47, I can’t understand the caveat being raised. Is it that provided words are already known to the network then it can do role-filler binding? What kind of network and how was it used? These questions need to be answered at appropriate length, either directly or indirectly, for the whole of the citations given in this paragraph since architecture and training regimes could differ dramatically and since this seems to be the springboard for this research.

It might be good to include references to others who have performed decoding analyses on artificial neural networks, for example, see the work and references in this preprint: https://doi.org/10.1101/626374 — it has a variety of papers that will help situate this part of the work into the wider literature on neural networks’ internal states.

Lines 253-4 really are the results here and I would love to see more on what this means. You need to do a bit more justice to your results and discuss them more. I feel like a broken record to keep repeating this, but it’s important and I struggle to judge this work without the authors writing a more self-contained piece.

Lines 255-258 describe a difference between your work and those of Puebla et al. I would have liked more here, expounding on the meanings of these differences in findings. Do you have any hypothesis or even overarching theory as to why an external memory is important here?

Experimental design

The paper is clearly written in most places and provides information on the experiments carried out but not an appropriate amount to situate this fully into the literature. Citing relevant papers in the literature is the first step, which seems to be attempted to a point, but the authors could also explain how their work is situated and what it builds on more. What theoretical account or position on schemas does this modelling work espouse? And in the discussion: what do the results mean for the literature and schema theory as well as for connectionist modelling? To be very blunt, I am not convinced the authors have engaged enough with the literature they cite and/or appropriately cite it.

The experiments' rationale is explained since the authors wish to show that it is possible to train networks to perform dynamic role-filler binding. However, as I mentioned already there needs to be more information to explain why these questions are important to psychology, connectionism, etc.

The supplementals could be, and maybe should be, incorporated into the main text and the code repository also linked to from the main manuscript.

Validity of the findings

I would be better placed to comment on this if the manuscript had more context here and explained the important details relevant to situating these findings into the wider field.

·

Basic reporting

This paper reports artificial neural network simulation results exploring various network architectures in search of one that can learn to process encoded “word” sequences in such a way so as to allow the network to correctly identify the word that fills a particular event schema role that is queried after sequence presentation. Importantly, a network is sought which can learn this task (1) without roles being explicitly labeled in the word sequence and (2) in a way that generalizes to arbitrary role-filler combinations.

For the most part, the article is clear and easy to read. The simulations appear to have been performed with competence, and the simulation results reveal salient differences in the capabilities of various network architectures to master the targeted role-filler binding task from training data.

There are three primary ways in which the presentation of this work could be improved, clarifying its contribution to the relevant literatures.

First, further technical detail is needed concerning the simulations that were performed, allowing for replication of the work. Some of the important missing details are outlined below.

Second, the document needs to be extremely explicit about what the authors view as sufficient evidence that a network can “perform role-filler binding”. I suspect that some readers will firmly reject the provided interpretation of the simulation results because they are committed to a conception of “role-filler binding” that is defined by some formal computational properties rather than defined by behavioral capabilities. The manuscript needs an explicit statement approximately of the form, “We interpret any system exhibiting behavior with properties X, Y, & Z as capable of performing general role-filler binding.” Ideally, this statement would be supported by citations to publications that make use of similar operationalizations of the notion of role-filler binding. (If this capability is thought to be exercised by humans during language understanding, there must be some aspects of human linguistic behavior that justify ascribing this capability to people.) I expect that being explicit about this matter will make transparent the degree to which residual contention is merely the result of different theoretical commitments concerning the defining characteristics of “role-filler binding”.

Third, the article should more thoroughly highlight the most focal feature of this work — the ability to perform using completely novel fillers. The current draft seems to put emphasis on the lack of explicit role labels, discussing the capability to deal with arbitrary fillers in a somewhat secondary way. While both of these features are important to the reported work, the ability to deal with completely novel fillers in a learning network seems to provide the focal contribution to the literature. (This emphasis does appear in some portions of the paper, such as “This occurs with words that were never seen during training, and with inputs that do not explicitly encode role-filler pairs.” [lines 48-50]) Consider, for example, that the Sentence Gestalt (SG) network (St. John & McClelland, 1990) received input word sequences without role labels, but it was able to answer role queries very similar to the ones that were used in the simulations reported here. Other work with simple recurrent networks (Elman, 1990) did not involve explicitly answering role queries, but still discussed networks that learned role bindings in a manner that improved prediction tasks. Thus, the focal contribution of this work involves generalization to arbitrary filters. Of course, there has been previously published research involving systems that can handle arbitrary fillers but require role labels. (One important example of such research, which is not cited by this manuscript, involves the work on Holographic Reduced Representations (Plate, 1995), including their neural network implementation in the Semantic Pointer Architecture (Eliasmith et al., 2012).) Such past systems have not acquired the capability to use arbitrary filters in the context of learning, however. Thus, I would suggest, the focal contribution of the work reported in this article involves the ability to learn to process sequences in a manner that generalizes to completely novel fillers. This feature should be more clearly highlighted.

Experimental design

As previously mentioned, there are a variety of technical details that are important for understanding (and replicating) the reported work which have been elided in the current draft. Some of these details appear in the supplementary materials but would be appropriate for the body of the paper. For example, the information in the sections on “Epoch Sizes” and “Network Details” in the supplement should largely appear in the manuscript, proper. The use of “nonsense word” pads should also be mentioned in the primary manuscript, and further detail concerning the number and frequency of such padding words should be included. Making changes of this kind will, for example, make Figure 2 more clear, as the meaning of “epoch” will be made transparent. The number of filler vectors used (rather than describing them simply as a finite set [line 96]) would be welcome in the body of the paper. Other important missing technical details include:

* From what distribution were the 50-dimensional word vectors drawn [line 87]? Were they binary vectors? Were the elements selected independently and uniformly at random over some range? This distribution is important. For example, the Fast Weights architecture could benefit from binary representations.

* The modifications to the NTM should be further justified. For example, the paper states that, “We remove these links because role-filler binding should not require the network to maintain links between adjacent buffer slots” [lines 133-134]. Do such links get in the way of good performance? Do they slow training? While such links might seem clearly unnecessary, removing them raises the question of whether or not they are detrimental to the task at hand.

* The decoding analysis [lines 151-157] is unclear. For example, was the ridge regression performed including memory state vectors that appeared in the middle of processing an input sequence? Was there essentially one regression per potential filler or one regression per dimension in the 50-dimensional word vector space? This technique is an interesting contribution of the paper, but it needs to be more clearly explained.

Validity of the findings

As previously discussed, some readers might reject the offered interpretation of the results due to differing theoretical commitments concerning what “role-filler binding” is. This submission would benefit from an explicit statement concerning the kinds of behaviors that are posited to require a role-filler binding capability.

While the reported findings are valuable, by themselves, the provided efforts to explain these findings are somewhat unsatisfying. Some of the results are explained by the claim that, “Previous work found that neural networks perform quite poorly when the distribution of the training set differs from that of the test set …” [lines 200-203]. This doesn’t seem like a good characterization of the issue. The fact that the training and test distributions differed isn’t sufficient to explain the results. Consider, for example, training using “Unlimited Filler Training” but testing on a small subset of the fillers that appeared during training (or a small set of novel fillers). In this case, the training and testing distributions differ, but it is very likely that good test set performance will be found. It is more likely that generalization performance is being driven by specific properties of the training set distribution, as is suggested by the notion of “sufficient diversity” in “This suggests that networks require a sufficient diversity of training examples to learn relational structure, rather than learning non-generalizable statistical correlations” [lines 258-259; see also lines 276-279].

Perhaps a better explanation of the reported results involves the ways in which some architectures are insensitive to spurious correlations in the training set. The literature on artificial neural networks is full of examples in which combinatoric generalization is thwarted by spurious correlations. Consider the reported “Robustness Tests”. In these analyses, the training set is designed to contain perfect anti-correlations between particular roles and particular fillers, and the networks are expected to ignore these anti-correlations in order to generalize. In the case of “Limited Filler Training”, the training set is designed so that only certain combinations of 50-dimensional word vector elements are allowed as outputs, with other vectors never appearing during training. A network that is able to learn the statistical distribution of output vectors would recognize that other combinations of output elements are never appropriate. This poses few problems when “previously seen” vectors are used during testing, but any network sensitive to these “spurious correlations” will clearly have problems if “previously unseen” filler vectors are used. From this perspective, the network architectures that succeed, in this paper, are those that contain a mechanism to retain filler vectors, as they were presented to the network, without distorting them based on the learned expected distribution of such vectors. Fast weights change in essentially the same way in response to a given input vector regardless of the past distribution of such input vectors. The reduced NTM contains explicit registers for maintaining arbitrary vectors. From this perspective, the inclusion of mechanisms that are insensitive to spurious correlations — they do not learn statistical regularities that are present in the training examples — are important for role-filler binding that generalizes in a combinatoric way.

The literature contains a variety of approaches to turn a blind eye to spurious (and other) correlations. Some mechanisms, like Holographic Reduced Representations (Plate, 1995; Eliasmith et al., 2012), don’t use any form of statistical learning, supporting generalization by design. Some mechanisms have involved fast weight learning in the context of highly sparse representations, such as some models of the hippocampus (Norman & O’Reilly, 2003). Some mechanisms have involved structured architectures with multiplicative gating (O’Reilly & Frank, 2006; Kriete & Noelle, 2011; Kriete et al., 2013). The approaches share the common goal of improving combinatoric generalization by limiting the effects of statistical learning on internal representations.

This submission takes a step beyond these previous efforts by (1) focusing very explicitly on binding, (2) focusing very explicitly on a lack of role labels, and, most importantly, (3) generalization to truly arbitrary filler vectors in a learning framework. Understanding the reported findings in terms of mechanisms for avoiding the learning of spurious correlations situates this work within the broader network learning literature.

Additional comments

The zero results for “Previously Unseen Filters” in Figure 3 makes the diagram somewhat confusing, with the blank spaces where graph bars are expected being disorienting. Perhaps the “Previously Unseen Filters” results should simply be removed from the graph, with the caption explaining that all values were zero.

In summary, this is work that is worthy of publication. My only recommendations are in terms of strengthening the presentation of the work.

Reviewer 3 ·

Basic reporting

I was not a reviewer on the previous submission of this ms., and so am commenting on the paper as a reviewer seeing the paper for the first time.

In short, I don’t think the simulations reported in the paper justify the paper’s main claims. I try to elaborate on this point below. I think the paper could be published if the claims are refined, and the relevant literature is addressed.

First, while this paper cites a bunch of papers on previous work on representing role-bindings in neural networks (I believe suggested by a previous reviewer), these papers haven’t been very carefully read, and it shows. For one thing, the claim that previous models required labelled role-bindings in the input is patently wrong for at least one of the cited models (Doumas et al.). Also, the cited St.John model uses the exact same input and test procedure as the one used in the current paper, so it’s odd to say that version uses labeled role-filler bindings while the previous one doesn’t. I think reading the literature would be really helpful in framing the current paper.

Second, the kinds of role-bindings that the current model could learn in principle are not dynamic and should be contrasted to dynamic role-filler bindings. Again, there’s a long literature on this topic, some of which is mentioned by the previous reviewer (e.g., Doumas & Hummel, 2005, 2012; Hummel, 2012). Reading that work would be helpful.

Third, the result in the paper seems very likely the result of the model learning to ignore certain bits of information when asking certain questions, rather than literal role-binding. This result was also reported in the cited Puebla et al. paper. In that paper, the finding was that when all fillers appeared in all roles with roughly similar frequency—as actually occurs in the current model as the small distributed encoding space of possible fillers (50 nodes) and large numbers of generated fillers almost assures all of the 50 nodes appeared in each role (though not in all possible combinations); this process is called spanning the input space and the practice as it relates to role-filler binding is covered in the previous literature suggested by the previous reviewer (Doumas & Hummel, 2012)—the model effectively learned to ignore context when answering specific questions. For example, if trained that a set of 4 people go to four coffee shops and order 4 different things and all combinations are roughly equally present in training, the model learns to return a copy of the y item in the sentence “x ordered y” in response to the question, “what was ordered?”. The problem is that then the model loses the capacity to learn any systematic context effects. If the model is then trained that John always orders cake when he goes to the coffee shop (“John went to the coffee shop” “John ordered cake”), the model is no more likely to return cake as, say, carrots when given the sentence “John went to the coffee shop” and asked “what did John order”. In other words, training as the done in the current paper, makes the model insensitive to context, while training with specific systematic statistical regularities (as in the other Puebla et al. sims) makes the model fail at returning the correct items when the statistically regular role-bindings are violated.

Finally, there’s a brief mention of a test where the sentences were mixed up (violated the regular story structure), and the model failed to correctly answer role-binding questions. Am I reading that right? Does that mean that when the model was given a story like, “John left the coffee shop; John listened to an MC; John left the coffee shop; John ordered coffee”, and asked what did John order, the model did not return coffee? If so, that’s a clear indication of failing to perform role-binding.

As a minor point, the story schema used was very regular and followed a specific structure where a basic story could be augmented, but the basic structure was always consistent. Did that have any effect on the results, perhaps?

Experimental design

No comment.

Validity of the findings

See above.

---

## Round 0.2 · Major Revisions

The three reviewers have gone through your revised manuscript. Although improvements were observed and acknowledged, some concerns remain that prevent publication in the current form, and additional improvements were requested. We therefore invite you to revise and resubmit your manuscript, addressing the points raised. Please highlight all changes in the manuscript text file.

Reviewer 1 ·

Basic reporting

I applaud the authors on refining their points and improving their manuscript. I think it's clearly written and the figures are fine — although I would suggest figures 3, 4 and 5 really need to have a larger font size to be readable.

Given their edits, clarifications, and expositions, I am pleased to say it has helped me get to grips more with the area and I suggest they look into the following papers I linked to below. Literature review is really important, so I am glad the steps being taken towards carrying it out are happening.

This body of work by Martin and Doumas seems particularly important for the authors to get grips with and I believe will help them engage with the wider context of their subfield/work. I list these just as pointers, I do not expect them to cite them (all) necessarily. However, I want to make sure the authors are aware of this body of work because it is modern, reflecting the current state of the field (which is important to prove relevance and facilitate communication and I believe the more historical papers like ones from the 1990s are covered, as well as the 2000s). I will leave it to the authors, and trust them fully, to connect these papers to their own theories and experiments as they see fit.

Martin, A. E. (2020). A Compositional Neural Architecture for Language. Journal of Cognitive Neuroscience.

Martin, A. E. & Doumas, L. A. A. (2019). Tensors and compositionality in neural systems. Philosophical Transactions of the Royal Society B: Biological Sciences.

Martin, A. E. & Doumas, L. A. A. (2019). Predicate learning in neural systems: Using oscillations to discover latent structure. Current Opinion in Behavioral Sciences.

Doumas, L. A. A., & Martin, A. E. (2018). Learning structured representations from experience. Psychology of Learning and Motivation, 69, 165-203.

Martin, A. E., & Doumas, L. A. A. (2017). A mechanism for the cortical computation of hierarchical linguistic structure. PLoS Biology, 15(3), e2000663

Experimental design

In line with the Aims & Scope of the journal (1 and 2) and with aiding understanding, each experimental manipulation should be clearly and carefully paralleled with something biological organisms do, in this case people. To really clarify and bring home the points the authors wish to make. I think the manuscript is really taking shape and this is perhaps the final important step outside addressing some further gaps in literature review I mention above.

On a less vital but still important note, I think it’s good for the author to engage with relevant papers on theory and use of neural networks. For example, when the authors state: "No networks consistently succeed in performing role-filler binding with shuffled story test sets.” Can the authors perhaps relate their results to these: https://arxiv.org/abs/1611.03530 — where networks are able to learn with shuffled labels? Again as with above, references to other articles should be taken as pointers to engage with the literature and not as requests for citation necessarily.

Validity of the findings

The discussion should include clear and numerous pointers to past work and how the current work adds to our understanding of the human organism and human behaviour: the explicit biological phenomena and capacities of interest here. Based on that the authors might benefit from looking at the papers I linked them to above and doing a bit more interlacing of their literature review with their own findings. Zooming out in other words and correctly contextualising their word within the social and biological sciences of linguistics and psychology — if their work is meant to be seen as a cognitive computational model of such phenomena the similarities and differences should pop out to the reader.

Bearing in mind the Aims & Scope of the journal (1 and 2 — as I mentioned previously), it's of importance to take these modelling results and tie them back in to the theories and hypothesis being tested and explored. This is important for readers to understand the contribution here and appreciate the work the authors have carried out. To really see why this is vital and why making their theory explicit will pay dividends, see, e.g., https://doi.org/10.31234/osf.io/rybh9 which explains in detail.

·

Basic reporting

This paper reports artificial neural network simulation results exploring various network architectures in search of one that can learn to process encoded “word” sequences in such a way so as to allow the network to correctly identify the word that fills a particular event schema role that is queried after sequence presentation. Importantly, a network is sought which can learn this task (1) without roles being explicitly labeled in the word sequence and (2) in a way that generalizes to arbitrary role-filler combinations (i.e., exhibits combinatoric generalization).

This revision is substantially improved in comparison to the previous manuscript. Additional technical details, needed for both understanding and replication, have been provided. The authors are more explicit about what is meant by “role-filler binding”, which is critical given the existence of multiple perspectives on this notion. The revision does a better job of situating the reported work in relation to previously published results. There appears to be a greater appreciation for the importance of the statistical structure of training patterns in networks of this kind, as well as the ways in which such structure can thwart combinatoric generalization, but the document still lacks a clear discussion of how the reported results could be largely explained by the differences in sensitivity to statistical structure across the network architectures. Indeed, the addition of a simulation experiment intended to show some retention of the statistical structure of training patterns, while welcome, seems to miss the fundamental contention between sensitivity to statistical regularities and the target goal of perfectly general role-filler binding. This last issue is discussed, below.

Experimental design

As previously mentioned, important details concerning the discussed simulations, which were elided in the previous draft, have been included in this revision.

Some of these details reveal potential imbalances across compared training regimens. For example, the variation in filler vocabulary sizes could influence the reported results, even beyond chance performance rates, given that word outputs were determined by a “code book” look-up over vocabulary items. Still, it is unlikely that any effects arising from these imbalances fully explain the important reported differences in performance.

The added analysis, intended to demonstrate sensitivity to the statistical structure of training patterns even in the network architectures that exhibited the strongest combinatoric generalization, is problematic. While the analysis is appropriate for showing that some sensitivity to the training distribution was acquired, the learned statistical structure is trivial, and focusing on such trivial learning suggests a lack of understanding of why an analysis of this kind is important. The focal conceptual issue, here, is discussed below.

Validity of the findings

As in the previous draft, the reported findings are certainly valuable, but the paper is still somewhat unclear about why the architectures that performed particularly well did so. The reader is told that these results indicate that an “external memory” is important for generalization, but the reader is not led to understand what distinguishes an “external memory” from other network structures. It is quite likely that the benefit of an “external memory” arises from its insensitivity to the statistical structure of training patterns. As suggested in a previous review:

“From this perspective, the network architectures that succeed, in this paper, are those that contain a mechanism to retain filler vectors, as they were presented to the network, without distorting them based on the learned expected distribution of such vectors. Fast weights change in essentially the same way in response to a given input vector regardless of the past distribution of such input vectors. The reduced NTM contains explicit registers for maintaining arbitrary vectors. From this perspective, the inclusion of mechanisms that are insensitive to spurious correlations — they do not learn statistical regularities that are present in the training examples — are important for role-filler binding that generalizes in a combinatoric way.”

The added experiment intended to show that some statistical structure was learned only demonstrates that the most obvious properties of training patterns (the mean values of the output elements) are retained. This distracts the reader from the fundamental opposition between statistical learning and truly general role-filler binding (particularly with novel filler vectors). A network that learns from training patterns that Anna is always the emcee will exhibit that knowledge in its performance, and it will, thus, have a more difficult time when presented with inputs that indicate that Belle is the emcee. If a network shows no difference in performance across fillers, even when training data reflects biases for certain roles to take on certain fillers, then the network is not exhibiting sensitivity to the correlations inherent in the network’s experiences. There appears to be an essential trade-off, here.

This is one reason why having an explicit criterion for the presence of role-filler binding capability is important. There is a “competence” view (embraced in this manuscript) that suggests that role-filler binding only exists when all fillers are managed equally well (e.g., perfectly). But this approach to role-filler binding runs counter to observed human behavior. People will have more difficulty remembering, and respond slower, to a stimulus like “the criminal arrested the officer” than to a stimulus like “the officer arrested the criminal”. Now, people can parse and understand the first of these two sentences, demonstrating some capability to overcome the statistical regularities of past experiences, but they do not perform equally well regardless of the nature of the fillers. This leads to a “performance” view of role-filler binding, which allows for sensitivity to statistical structure by slightly degrading combinatoric generalization.

The newly added experiment uses “ambiguous” inputs for an important reason. If the identity of the poet is not mentioned in the input, but Carter was almost always the poet during training, then presenting a query about the identity of the poet should show a bias toward a Carter response, if the “performance” perspective is taken. As presented in this draft, using “ambiguous” inputs seems superfluous, as all that is being demonstrated is the learning of the mean output values during training — not any correlations between roles and fillers. (The Sentence Gestalt network demonstrated this kind of statistical learning.) Thus, it seems as if the design of this added experiment is based on some misunderstanding of the importance of the trade-off between sensitivity to training distributions and clean combinatoric generalization. A better demonstration would be to bias the training data (e.g., the poet is usually Carter) and compare the cosine distance to the nearest vocabulary item when tested on an input indicating that Belle was the poet versus Carter. If the network correctly identifies Belle as the poet when she is, but the output vector is farther from the Belle vector in comparison to the Carter case, then the network is demonstrating combinatoric generalization while exhibiting some sensitivity to the training distribution.

I suspect that a reasonable balance in this trade-off can be more easily had in networks that have some part that is insensitive to training statistics (i.e., the “external memory”) and some part of the network that learns training correlations (e.g., other weights).

Additional comments

In summary, this is work that is generally worthy of publication, though a better experimental design to show sensitivity to training distributions would be welcome. The paper could also be further strengthened by discussing the trade-off between learning correlations and avoiding the trap of “spurious correlations” that hinder robust combinatoric generalization.

Reviewer 3 ·

Basic reporting

See below.

Experimental design

See below.

Validity of the findings

See below.

Additional comments

Review of “Learning to perform role-binding…” for PeerJ

I reviewed a previous version of this paper. After reading this version many of my previous concerns still stand.

One problem with the paper is that while the ms. starts by describing dynamic role-filler binding, it then presents a definition of role-binding that is definitely not dynamic. It’s fine to define role-filler binding in a new way, but it should be clearly contrasted with the way it is conventionally defined. Especially, when initial text refers to conventional discussion.

On p. 2 the reference to the definition of role-filler binding by Holyoak and Hummel is incomplete. Hummel and Holyoak also argued that role-filler binding must be dynamic. It is misleading to give the current definition of role-biding following a discussion of Holyoak and Hummel, when half of Holyoak and Hummel’s definition is left out. The current definition is an odd one to have been motivated to by Holyoak and Hummel.

The discussion of prior would once again benefit from more careful reading of the cited work. First off, I don’t think anyone has claimed (and certainly not the papers cited in the first paragraph of the Related work section) that: “Connectionist models were traditionally thought to be unable to represent these symbolic relations, because of an ability to independently represent roles and fillers”. I think “inability” is intended rather than “ability”, but either way the papers don’t argue that traditional connectionist models can’t independently represent roles and fillers (that’s easy, just have one node for the role and one for the filler). What they argue is that traditional connectionist networks can’t bind independent representations of roles to fillers dynamically. That’s a major point in this area.

Second, the discussion of Doumas et al. (2008) is again wrong. First, the sequence of comparisons are not hand-crafted (except in the first sim, then second sim shows they don’t have to be), and the comparison is only important for learning the relations in the first place, not for binding. Second, the binding signal emerges as a function of the architecture (layers of laterally inhibititive units and separate banks). If the architeture means hand-crafted, then your model is hand-crafted too (see organisation, learning rules, etc.).

Finally, I think my original points about the limitations of the current simulations still stand. I wrote:
“Third, the result in the paper seems very likely the result of the model learning to ignore certain bits of information when asking certain questions, rather than literal role-binding. This result was also reported in the cited Puebla et al. paper. In that paper, the finding was that when all fillers appeared in all roles with roughly similar frequency—as actually occurs in the current model as the small distributed encoding space of possible fillers (50 nodes) and large numbers of generated fillers almost assures all of the 50 nodes appeared in each role (though not in all possible combinations); this process is called spanning the input space and the practice as it relates to role-filler binding is covered in the previous literature suggested by the previous reviewer (Doumas & Hummel, 2012)—the model effectively learned to ignore context when answering specific questions. For example, if trained that a set of 4 people go to four coffee shops and order 4 different things and all combinations are roughly equally present in training, the model learns to return a copy of the y item in the sentence “x ordered y” in response to the question, “what was ordered?”. The problem is that then the model loses the capacity to learn any systematic context effects. If the model is then trained that John always orders cake when he goes to the coffee shop (“John went to the coffee shop” “John ordered cake”), the model is no more likely to return cake as, say, carrots when given the sentence “John went to the coffee shop” and asked “what did John order”. In other words, training as the done in the current paper, makes the model insensitive to context, while training with specific systematic statistical regularities (as in the other Puebla et al. sims) makes the model fail at returning the correct items when the statistically regular role-bindings are violated.”

I think that the correlation violation results are interesting. There are clear differences between the correlation violation used in Puebla et al. and in the current paper, however. More importantly, it’s not clear why the particular test of the network’s ability to retain statistical information was used rather than a simpler one. Based on the Puebla et al. tests, the test that should be run is as follows: the model is trained on a bunch of inputs where John orders cake (and only cake) at the coffee shop, and other people order other stuff. The network is trained to a point where when told John orders pie at the coffee shop and it returns pie to the question what did John order. Then the network is told John was at the coffee shop and asked what John ordered. Does it then return cake, or has this information been lost. Whey wasn’t this simple simulation used rather than the present one?

Second, the model succeeds in binding “novel” roles to fillers only in the unlimited training, when there are no systematic fillers used. That seems to suggest that the generalisation occurs when the model learns to ignore the filler information. I am skeptical that the unlimited training model could perform as it did if there were systematic statistical properties of the fillers of specific roles.

Also, I think it’s rather misleading to claim that the model is using unseen fillers when it used a pool of input that were clearly spanned (using only 50 inputs to code the fillers means likely each unit was the answer to each question at least once during training; again, see the spanning the input space point I made in my previous review). In the context of the literature on role-filler binding, unseen inputs mean units not used during training, not unseen combinations of units. The claim about novel fillers should be amended to something like “novel combinations of previously trained units”.

---

## Round 0.3 · accepted · Accept

Thank you for the revised manuscript and response letter. I am pleased to inform you that your manuscript "Learning to perform role-filler binding with schematic knowledge" has been accepted for publication.

·

Basic reporting

This is the second submitted revision of a paper that reports artificial neural network simulation results which explore various network architectures in search of one that can learn to process encoded “word” sequences in such a way so as to allow the network to correctly identify the word that fills a particular event schema role that is queried after sequence presentation. Importantly, a network is sought which can learn this task (1) without roles being explicitly labeled in the word sequence and (2) in a way that generalizes to arbitrary role-filler combinations (i.e., exhibits combinatoric generalization).

As described below, this draft is greatly improved in comparison to the previous revision. In the judgement of this reviewer, this manuscript is worthy of publication, with any remaining contentious issues addressed through conversations between scholars in the literature.

Experimental design

The reported simulation experiments are well described. Previously elided details are now present. Concerns over imbalances in training set sizes have been explicitly discussed, adequately addressing worries about confounds.

Most importantly, the provided analyses that are intended to demonstrate sensitivity to the statistical structure of sets of training patterns, even in some of the network architectures that exhibited the strongest combinatoric generalization, are substantially improved. While some might argue for more concise methods to demonstrate sensitivity to “spurious correlations”, the logic behind the experimental designs used in this revision is certainly sound, and the results are compelling. Furthermore, the justifications for the designs reflect a deeper understanding of the theoretical issues being confronted through the provided analyses.

Validity of the findings

In a review of a previous draft of this manuscript, it was noted that the article’s discussion of the key differences between architectures likely responsible for differences in behavior failed to succinctly reflect important general insights into the causes of the reported results. That review suggested:

“From this perspective, the network architectures that succeed, in this paper, are those that contain a mechanism to retain filler vectors, as they were presented to the network, without distorting them based on the learned expected distribution of such vectors. Fast weights change in essentially the same way in response to a given input vector regardless of the past distribution of such input vectors. The reduced NTM contains explicit registers for maintaining arbitrary vectors. From this perspective, the inclusion of mechanisms that are insensitive to spurious correlations — they do not learn statistical regularities that are present in the training examples — are important for role-filler binding that generalizes in a combinatoric way.”

This critique has been fully addressed in the current revision. The issues are clearly discussed, and they are further explored in the context of the new simulations focusing on the learning of statistical relationships.

In the judgment of this reviewer, the phenomena of interest have been adequately demonstrated and discussed. While the terms used to describe these phenomena are appropriately defined and associated with their use in the literature, it is possible that other readers will still take issue with the reported findings due only to differences in preferred meanings of terms. The existence of conflicting viewpoints in the literature likely makes this problem impossible to avoid. In this reviewer’s opinion, the current revision is sufficiently clear with regard to the project’s goals and the validity of the associated findings in the context of those goals.

Additional comments

In summary, this is work that is worthy of publication. Previously raised concerns have been addressed in this revision, and no new problems are seen to have been introduced.